# Genomic Instability and Epigenetic Changes during Aging

**DOI:** 10.3390/ijms241814279

**Published:** 2023-09-19

**Authors:** Lucía López-Gil, Amparo Pascual-Ahuir, Markus Proft

**Affiliations:** 1Department of Biotechnology, Instituto de Biología Molecular y Celular de Plantas, Universitat Politècnica de València, Ingeniero Fausto Elio s/n, 46022 Valencia, Spain; luciaupv22@gmail.com; 2Department of Molecular and Cellular Pathology and Therapy, Instituto de Biomedicina de Valencia IBV-CSIC, Consejo Superior de Investigaciones Científicas CSIC, Jaime Roig 11, 46010 Valencia, Spain

**Keywords:** aging, genomic instability, epigenetics, chromatin, histones

## Abstract

Aging is considered the deterioration of physiological functions along with an increased mortality rate. This scientific review focuses on the central importance of genomic instability during the aging process, encompassing a range of cellular and molecular changes that occur with advancing age. In particular, this revision addresses the genetic and epigenetic alterations that contribute to genomic instability, such as telomere shortening, DNA damage accumulation, and decreased DNA repair capacity. Furthermore, the review explores the epigenetic changes that occur with aging, including modifications to histones, DNA methylation patterns, and the role of non-coding RNAs. Finally, the review discusses the organization of chromatin and its contribution to genomic instability, including heterochromatin loss, chromatin remodeling, and changes in nucleosome and histone abundance. In conclusion, this review highlights the fundamental role that genomic instability plays in the aging process and underscores the need for continued research into these complex biological mechanisms.

## 1. Introduction

Current demographic aging, or the trend of populations becoming older, is the result of declining birth rates and increasing life expectancy. This phenomenon poses a number of challenges for societies, including increased demand for healthcare and social services, a shrinking workforce, rising pension and healthcare costs or slower economic growth [1,2]. As of 2022, 6% of Spain’s population is aged 80 and over, accounting for approximately 2.8 million people [3]. Thus, it is highly relevant to achieve a better understanding of the mechanisms underlying aging in order to progress toward a healthier elderly population.

Aging refers to the process of biological deterioration that occurs in an organism over time. It is characterized by a decline in physiological functions a decrease in adaptability to environmental stresses and is accompanied by an increased mortality rate and an increased risk of developing a variety of diseases such as cancer, inflammation, cardiovascular diseases, and neurodegenerative disorders [4]. At the cellular level, aging is associated with a range of molecular and biochemical changes. Currently, twelve hallmarks of aging have been defined: genomic instability, telomere shortening, epigenetic alterations, loss of proteostasis, disabled macroautophagy, deregulated nutrient sensing, mitochondrial dysfunction, cellular senescence, stem cell exhaustion, altered intercellular communication, chronic inflammation and dysbiosis [5]. All of these attributes are highly intertwined; thus, the modulation of one affects others (Figure 1).

Aging is influenced by established genetic factors and environmental factors. Among these, epigenetics account for the majority of the observed variation, as proved through twin studies. The comparison of monozygotic twins’ epigenetic marks revealed that young twins have similar patterns, whereas aged twins show more divergence, indicating that the epigenetic variations observed in old people are a result of the adaptation to environmental, non-heritable influences [6,7,8]. Epigenetic changes imply modifications in gene expression without alterations of the underlying DNA sequence but through other modifications mainly affecting chromatin [9]. These modifications include telomere alterations, post-translational modifications of histones such as methylation, acetylation, phosphorylation or ubiquitination, reorganization of chromatin through the alteration of heterochromatin extension, nucleosome occupancy or chromatin remodeling, changes in histone variants, altered non-coding RNA expression and modifications of the epitranscriptome.

Alongside these traits, physiological drivers of the aging process and age-related diseases, such as mitochondrial function, proteostatic and autophagic capacity or nutrient sensing, have been extensively studied, and the reader is referred to excellent recent revisions covering those fields [10,11,12,13,14]. This review will summarize the phenomenon of genome instability associated with aging through several mechanisms, which have been revealed thanks to studies performed in a wide variety of model organisms, highlighting the use of *Saccharomyces cerevisiae*, *Caenorhabditis elegans*, and *Drosophila melanogaster*, along with mice and human cells.

## 2. Telomere Shortening

Telomeres are specific regions located at the ends of chromosomes whose role is to protect the genetic information contained within the chromosome from gradual degradation that occurs after each cell division. Thus, they play a critical role in the maintenance of genome integrity. The degradation of telomeres is gradual and intrinsic to the normal lifespan of cells. After continuous divisions, once the telomeric region protecting the ends of the chromosomes reaches a critical length, division stops. The cell activates genetic programs such as replicative senescence due to telomere shortening, and the accumulation of senescent cells can accelerate tissue and organismal aging. In fact, high numbers of senescent cells have been found in aged individuals [15].

Very interestingly, telomere shortening is functionally connected to mitochondrial performance and vice versa. During cellular senescence, malfunctioning mitochondria generate reactive oxygen species (ROS), which, in turn, cause functional impairment in proteins participating in oxidative phosphorylation (OXPHOS) [16]. Mitochondrial dysfunction can lead to telomere attrition [17,18], and mitochondrial ROS scavengers are able to decelerate telomere shortening [19]. Owing to the higher proportion of vulnerable components within telomeres, they are more susceptible to oxidative harm, accelerating telomere shortening [20]. Mitochondrial-generated ROS can freely diffuse into cell nuclei, serving as key mediators of communication between mitochondria and nuclei. In instances of mitochondrial dysfunction, telomeric harm ensues without concomitant nuclear DNA damage [21]. Patients afflicted with mitochondrial disorders exhibit shorter telomeres compared to their healthy counterparts, suggesting a plausible connection between mitochondrial dysfunction and telomere deterioration [20]. On the other hand, damage to telomeres can cause mitochondrial biosynthesis to be reprogrammed, resulting in mitochondrial dysfunction [22,23]. Telomere damage primarily impacts mitochondrial function through the activation of p53 and inhibition of the PGC-1α pathway, with p53 playing a crucial role in telomere attrition and DNA damage [24]. PGC-1α is a key regulator of mitochondrial biosynthesis and metabolism and serves as a pivotal link between telomere damage and mitochondrial dysfunction. Telomere shortening affects p53 and PGC-1α expression, ultimately leading to mitochondrial impairment. Recent research reveals that telomere DNA damage precedes mitochondrial metabolic disorders. Treatment with KML001, a telomere-targeting drug, induces cellular senescence and apoptosis by impairing telomere function, which activates the p53 signaling pathway. Crucially, p53 regulation can influence the expression of PGC-1α and NRF-1 to enhance mitochondrial function, including mitochondrial membrane potential, oxidative phosphorylation, mitochondrial respiration, glycolysis, and ATP synthesis [24]. These interactions between mitochondria and telomeres have significant implications for aging and disease [20].

Telomerases are RNA-dependent DNA polymerases, which are able to regenerate telomeric regions [25,26,27,28]. However, most somatic cells do not express this protein; thus, the progressive accumulation of shortened ends leads to the detention of the cell cycle [29]. The rate of telomere shortening is highly variable and is further enhanced at the organismal level by external factors such as stress, diet, or economic status [30,31], as well as cellular-level factors such as cell stress, oxidative damage or hormone signaling [32]. Specifically, oxidative stress has been shown to contribute to telomere alterations; for example, ROS derived from mitochondria have been shown to damage the telomeres and initiate senescence [33,34].

Generally, telomere shortening is a phenomenon that has been observed in most aged tissues [35,36,37] and in mouse models [38]. However, there are important differences between specific human tissues [39]. There are two possible endpoints to this phenomenon: apoptosis or senescence. Telomere attrition has been established to diminish the lifespan of both humans and mice [31,40], while the longevity of yeast, nematodes or flies was not affected by telomere length [41,42,43]. Importantly, the overexpression of telomerase or protection of telomere length has been shown to increase lifespan in mice [40,44,45]. Interestingly, it has also been observed that telomere shortening affects the self-renewal capacity of stem cells [46] and the viability of stem cell niches [47].

## 3. DNA Damage Accumulation

DNA damage is considered one of the main contributors to the degeneration that accompanies the aging process [48,49,50]. Despite being the fundamental blueprint for all genetic information, the genome is remarkably unstable. Genome instability is characterized by a tendency to undergo mutations, which are permanent, inheritable changes to the DNA sequence, such as base substitutions, insertions, deletions, chromosomal abnormalities, and retrotransposition. Mutations typically have harmful effects on function and are a significant contributor to cancer and genetic disease. However, in the germline, they also provide the basis for evolutionary change.

Mutations arise as a consequence of errors in replication or repair, often from DNA damage. In a broader sense, genome instability can refer to the propensity of DNA to undergo chemical modifications that alter its structure and function [51]. DNA damage can take many forms, including different types of breaks, nicks, gaps, abasic sites, adducts, and crosslinks, as well as other chemical modifications. Aberrant DNA structures, such as R-loops, G-quadruplexes, and persistent single-strand regions, as well as arrested intermediates in DNA transactions such as stalled transcription, replication, and recombination complexes, can also compromise DNA function and thus cause DNA damage. DNA lesions impair accurate replication, controlled transcription, and secure storage of genetic information. Unlike other biomolecules that can be remade based on instructions carried by corresponding genes, DNA integrity is maintained only through constant repair. Accordingly, genetic diseases leading to weakened DNA repair capacity or exposure to DNA-damaging radiation or chemotherapy accelerate the normal aging process [52,53,54].

Both genomic and mitochondrial DNA is continuously being damaged by endogenous genotoxins such as ROS or aldehydes, along with the actions of exogenous genotoxins such as chemical compounds in the environment, UV or X-rays. The perpetual unresolved damage altogether can result in genome instability and ultimately cause apoptosis [55] or activate senescence [56]. This occurs to avoid the replication of the accumulated damaged DNA; however, the age-dependent enrichment of senescent cells has been identified as a possible driver of aging [57]. To overcome this situation, cells have DNA damage repair systems with high efficiency, whose capacity, however, is compromised during aging [58,59]. It has been established that DNA damage accumulates with aging in mouse models and human tissues [60,61], which causes an increasing accumulation of somatic mutations during the aging process [62,63,64,65,66,67]. In fact, a recent systematic study across mammalian species revealed that somatic mutation rate is a dominant driver of aging [68].

There are several sources of intrinsic DNA damage, among which oxidative stress is one of the main driving forces. Oxidative stress is a detrimental state induced by the accumulation of ROS, mainly of mitochondrial origin. Their principal mechanism of action is through the oxidization of the guanine base of DNA, forming 8-hydroxydeoxyguanosine (8-OHdG). 8-OHdG can then pair with adenine instead of cytosine during DNA replication, leading to a mutation in the DNA sequence. These alterations can change the epigenetic patterns by damaging specific genomic regions, such as the CpG-rich promoter regions in mouse tissue [69]. Additionally, the ribosomal DNA (rDNA) locus is highly targeted by both endogenous and exogenous factors such as ROS and UV light, respectively, originally described in the yeast model [70]. This damage leads to genomic instability, a known trigger of aging, which ultimately triggers cellular senescence [71]. The highly repetitive rDNA locus is tightly controlled during normal aging to avoid copy number loss or the accumulation of potentially toxic extrachromosomal ribosomal DNA circles (ERCs) [72,73]. Although the function of ERCs as molecular drivers of aging is restricted to the yeast model and still not completely understood [74,75], instability of the rDNA genomic region is a general feature of aging cells [76,77]. In yeast, rDNA damage in the nucleolus is one of the two prominent death modes [78], and rDNA copy number is generally linked to longevity [79]. Recent studies in higher eukaryotes also suggest that rDNA repeats are particularly vulnerable to damage, and alterations in their copy number may contribute to aging and senescence [80,81,82,83]. Human rDNA is especially prone to replication stress due to complex tertiary DNA structures in the intergenic spacer of 45S rDNA [84]. Studies on aging mice and senescence in response to oncogene activation suggest an early involvement of the rDNA/nucleolus [85,86]. This alteration has also been documented in aged human samples, showing an increased instability due to the loss of rDNA [87].

Mitochondrial DNA (mtDNA) is greatly affected by aging too. Mitochondrial dysfunction caused by ROS-induced damage to the mitochondrial genome has been proposed as the primary cause of age-related mitochondrial dysfunction, according to studies in mice and human cells [88,89,90,91]. There is a resurgence of age-associated mutations due to the limited capacity of the DNA repair machinery in this organelle, the oxidative environment and the lack of protection provided by histones. However, the frequency of such mutations during natural aging is unclear [92,93], and the low frequency of mtDNA deletions and mutations in wild-type mice suggests that most somatic mtDNA mutations originate from replication errors during development [94]. Taken together, although an increase in mtDNA alterations is found in a wide variety of human tissues with age, its true impact is yet to be determined [5].

It is important to note that in many cases, DNA damage does not affect by itself the functionality and performance of the cell, but it has been postulated that it favors epigenetic modifications, which add to the aging onset [95]. Increasing evidence indicates that age-related alterations in epigenetics are largely influenced by DNA damage. During DNA repair processes, DNA methyltransferases are present at the damaged sites, and several chromatin remodeling factors play a role in the assembly of various repair mechanisms, which can result in the formation of epigenetic imprints upon lesion removal and restoration of the initial chromatin state. This has been documented in various models, from *Caenorhabditis* to mouse and human cells [96,97,98]. Thus, it is plausible that the constant occurrence of DNA damage and repair, resulting in the formation of epigenetic imprints, contributes to intercellular epigenetic heterogeneity and age-related decline in gene expression control [99]. Ultimately, the accumulation of DNA damage can induce unanticipated secondary stresses, such as for example proteostatic stress, affecting the structure and function of proteins [49]. This is exemplified by several findings, which link the deficiency in DNA damage repair mechanisms with age-related diseases based on the misfolding and aggregation of proteins, such as Alzheimer’s or Parkinson’s disease [100,101,102,103,104,105]. In turn, proteostatic defense mechanisms such as autophagy are activated and physiologically relevant upon increased DNA damage [106].

## 4. Decreased DNA Repair Capacity

Every living organism possesses resilient DNA damage repair (DDR) mechanisms to detect a great variety of DNA damage, pause the replication of its genome as necessary, notify for repair, and rectify or endure the numerous genetic injuries that arise regularly [107]. If DNA damage remains unrepaired for a prolonged time or is too severe to be fixed, it triggers signaling processes that result in different cellular outcomes, including senescence, which contributes to the process of aging. Its deficiency greatly affects the cell’s fate, sometimes having an even greater impact than the damage itself [50]. There are several mechanisms, such as Mismatch Repair (MMR), Base Excision Repair (BER), Nucleotide Excision Repair (NER) or Double Strand Break (DSB) repair and the interested reader is referred to excellent recent reviews on the molecular basis of DNA damage response pathways [108,109]. Here, we will focus on how a diminished DDR capacity accompanies and determines the aging process.

There are numerous reports identifying a decrease in DDR capacity along the aging process [59]. One example is the repair of DSBs, which constitutes one of the most damaging types of DNA lesions, leading to cell death if left unrepaired. The repair mechanism follows two major pathways: Homologous Recombination and Non-Homologous End Joining (NHEJ). In general lines, the lack of repair in this case has been reported to entail an increased frequency of genomic rearrangements due to the accumulation of mutations [58]. A diminished capacity of DSB repair machinery has been observed in peripheral lymphocytes of old patients [110], along with the establishment of low levels of repair in both old tissues [111] and senescent cells [112]. The decline of DDR capacity has additionally been reported to be tissue-specific. For example, NHEJ activity was tested in different tissue samples from young and aged mice, finding an overall decrease in activity. In parallel, an increase in alt-NHEJ, an alternative version of NHEJ that is more prone to have mutagenic effects, is upregulated [113]. Similar effects have been recently reported in the yeast model during replicative aging, in *Drosophila* and in mice, thus highlighting the conserved nature of DSB and other DNA repair defects in old cells [114,115,116,117]. Similarly, NER capacity is known to decline during nematode aging [118].

The correlation between aging and a decline in DDR capacity has been additionally supported by what is known as progeria syndromes, or premature aging disorders, which recapitulate many traits associated with aging in young individuals [119]. Inherited DNA repair pathway defects are associated with distinct genome instability syndromes characterized by developmental abnormalities, increased cancer incidence and features of accelerated aging. Progeroid syndromes, such as Hutchinson–Gilford, Werner, Cockayne, and XFE progeroid syndrome, are examples of diseases that cause dramatic accelerated aging and are linked to genome instability [120,121,122]. Genome stability is impaired in the Hutchinson–Gilford progeroid syndrome by mutations in the A-type nuclear lamina *LMNA* [123]. Null mutations in the *WRN* gene encoding a RECQ DNA helicase with important functions in DNA repair and recombination cause accelerated genome instability in Werner syndrome [124]. Different genetic alterations collectively cause an impairment of the nucleotide excision DNA repair system in Cockayne syndrome [125]. XFE progeroid syndrome results from mutations in *ERCC4/FANCQ*, causing reduced expression of *XPF-ERCC1*, a heterodimeric DNA repair endonuclease required for NER and the repair of some DNA DSBs [126]. Collectively, these syndromes illustrate the devastating impact of DNA damage on human health and are important experimental models for aging research [107]. Progeroid syndromes are caused by a variety of mutations affecting the expression or activity of key protein functions involved in DDR (in-depth revised in [50]).

Interestingly, the importance of an efficient DNA repair system in achieving longevity has been highlighted by a study carried out by Garagnani et al. in supercentenarians (110 years old) and semi-supercentenarians (105 years old). Through whole-genome sequencing and the comparison with young controls from the same geographic regions, they were able to identify an upregulated activity of DNA repair genes in the old cohort, along with lower mutational patterns than their young counterparts [127]. Accordingly, reinforcement of DDR capacity has beneficial effects on longevity, according to studies in different models. Comparing the aging process across rodent species, the sirtuin-regulated strength of DSB repair conditions longevity [128], while the overexpression of DDR genes in *Drosophila* has been shown to improve lifespan [129,130].

## 5. Histone Modification Changes

Histones undergo various chemical changes after being produced, known as post-translational modifications. These modifications mainly entail histone N-terminal tail methylation, acetylation, phosphorylation, ubiquitination, and ADP ribosylation [131,132]. The effects of histone modifications are varied; some of them have activating roles in transcription, while others promote gene silencing [133]. It is during aging that the accumulation of several modifications greatly disrupts the balance, leading to global changes in the genetic expression. However, it is important to take notice that histone alterations associated with aging follow very heterogeneous patterns even when comparing cells within the same tissue [6,134]. In other words, there is no uniform modification pattern [134,135], and it may be necessary to study them on a case-to-case basis. For example, Cheung et al. were able to identify a cell-type specific pattern of histone modification when analyzing blood samples from old individuals [6]. Of these modifications, lysine residues’ acetylation and methylation are the most extensively researched, with known effects on aging. Studies conducted in vivo and in vitro have revealed alterations in the levels of H3K9me3, H4K20me3, H3K27me3, and H3K9ac during the aging process. Here, we will briefly discuss some of the most relevant aspects of histone methylation and acetylation as determinants of the aging process in diverse cell models. For a more specialized and extensive review, please refer to Yi and Kim [136].

### 5.1. Histone Methylation

Histone methylation marks have been shown to have both activating and repressing roles depending on the residue affected. It has been established that methylation marks in H3K4 or H3K36 promote transcription, whereas methylation marks in H3K9 or H3K27 silence transcription [137]. Generally, aging entails the loss of repressing marks and the gain of activating marks in previously non-modified regions. All of this may contribute to the loss of heterochromatin and genomic stability or the deficiency observed in DDR mechanisms [134,138,139,140,141].

Several studies covering a wide variety of experimental models have been performed to better understand the role of histone methylation changes during aging. Some authors have been able to identify specific histone methylation patterns associated with aging in mice [142], *Drosophila melanogaster* [143] and *C. elegans* [144]. More in-depth, Cruz et al. described an increase in the levels of H3K4me3 in promoter regions and rDNA during aging in *S. cerevisiae*, linked to the activation of aging-related genes [145]. Similarly, characteristic H3K4me3 patterns have been identified in aging nematodes, which are required to execute gene expression in old individuals [146], and a reduction in excess H3K4me3 has been shown to increase longevity in the same model [147]. In line with these findings, the level of H3K4me3 and its catalyzing enzymes increases in the prefrontal cortex of a mouse model of Alzheimer’s disease, and treatment with an H3K4 HMT inhibitor improves prefrontal cortex function [148]. The H3K4me3 level increases with age in mouse hematopoietic stem cells [149], but in aged human neurons, there is a loss of H3K4me3 at some genes and a gain at others compared to young neurons [150,151]. Thus, although the influence of H3K4me3 on aging is repeatedly documented, it is context-dependent and requires further investigation.

Global changes affecting H3K27me3 have also been reported to affect the lifespan of several organisms. H3K27me3 is usually related to gene silencing and compacted heterochromatin [152]. Previous studies indicated a complex picture because there is a global loss of H3K27me3 in aged *C. elegans* and prematurely aged cells from Hutchinson–Gilford progeroid syndrome patients [153], while in killifish, mouse and human cells, global H3K27me3 increases with age [154,155]. Different histone lysine demethylases may work as positive regulators of lifespan in general and specifically in response to mitochondrial dysfunction across different species [144,156,157,158], suggesting that increased levels of H3K27me3 in genes involved in the mitochondrial unfolded protein response (UPRmt) are detrimental to lifespan [159]. On the other hand, high transcriptional variability correlated with age was found enriched in H3K27me3-marked genes [6]. Therefore, it will be crucial to determine the locus- and cell-type-specific roles of H3K27me3 in lifespan regulation to understand the effect of H3K27me3-modifying enzymes on aging. Finally, it has also been proposed that the age-associated altered H3K27me3 pattern may increase the risk of cancer in aged individuals. This derives from the interpretation that the altered methylation pattern causes an impairment in stem/progenitor cell differentiation, even causing a de-differentiation, that confers cells with traits such as death resistance, which are positively selected in the tissue microenvironment [160].

Additionally, adding to the complexity, the histone modifications H3K36me3 and H3K9me3 are involved in the aging process. In *S. cerevisiae* and *C. elegans*, a deficiency of H3K36me3 leads to a shorter lifespan, while the loss of H3K36me3 demethylase extends the yeast lifespan [161]. In *Drosophila*, the loss of H3K9me3 leads to intestinal stem cell aging [162], and in aged somatic tissues of *C. elegans* and *Drosophila*, the levels of H3K9me3 and HP1 decrease [163,164]. The expression of the H3K9me3 methyltransferase SUV39H1 is decreased during the aging of human and mouse hematopoietic stem cells, leading to a global reduction in H3K9 trimethylation and perturbed heterochromatin function [165].

### 5.2. Histone Acetylation

Histone acetylation is generally associated with active gene expression, as the modification reduces the tight interconnection between the histone octamer and the DNA. This process is mediated by two classes of enzymes: the histone acetyl-transferases (HATs), which incorporate the acetyl group and the histone deacetylases (HDACs), which remove acetyl groups from histone tail lysines [166,167]. Both HATs and HDACs play critical roles in establishing longevity [136].

Several acetylation marks have been extensively studied in the context of aging, among whichH3K9 acetylation has played a very important role. In general lines, it has been discovered that this mark decreases with age when studying liver samples from young and old rats [168]. Furthermore, acetylation patterns change globally at many H3K9ac domains in aged human prefrontal cortices [169]. Additionally, H4K16ac marks have been found to be increased during aging in both yeast [170] and human cells [171]. Moreover, H3K56 acetylation, known to play an essential role in promoting nucleosome assembly, genomic stability, transcription, and chromatin organization [172,173], has been linked to aging. In yeast, it has been observed that the level of H3K56ac decreases with age and that Rtt109, a major acetyltransferase for H3K56, is important for lifespan extension, but the role of H3K56-associated deacetylases Hst3 and Hst4 is less clear [174]. Thus, a delicate balance of acetylated H3K56 may be necessary for promoting longevity. Collectively, alterations affecting the acetylation patterns have been shown to be crucial regulators of homeostasis during aging and the preservation of genome stability [175].

Additionally, the role of HDACs, mainly the Sirtuin family of NAD^+^-dependent histone deacetylases, has been the focus of several aging studies [176,177]. Most notably, the absence of Sirt6 deacetylase, which targets H3K9ac, causes serious negative effects in metabolism and survival in mice [178] as it plays a key role in the regulation of cellular processes such as genomic stability [179]. Consequently, its overexpression led to increased lifespan in mice [180,181]. Another deacetylase, Sir2, the founding member of the sirtuin family originally discovered in yeast and ortholog of mammalian SIRT1, which targets H4K16ac and H3K56ac [182], has been proved to have a beneficial effect on longevity when overexpression studies were conducted in yeast [183,184,185], worms [186] and flies [187]. Interestingly, the brain-specific overexpression of SIRT1 in mice enhances lifespan [188]. Conversely, a decline in Sir2 activity has been associated with aged organisms, thus producing an increased H4K16ac mark level and consequent genomic instability along with histone loss in yeast [170]. In the same vein, reducing H4K16 acetylation by inactivating the corresponding HAT activity has been shown to improve lifespan in the case of the Sas2 HAT. Sas2 is the major H4K16 acetyltransferase that establishes boundaries between telomeres and euchromatin [189,190], and its deletion extends lifespan [170]. Other HATs have also been shown to be important for regulating longevity. In *Caenorhabditis*, decreased levels of H3K5 acetylation, which is associated with aging, can be suppressed via dietary restriction and accelerated via inhibiting the expression of the HAT CBP-1 [191]. In a recent study, manipulation of CBP was found to affect the lifespan and offspring production of the pea aphid, suggesting that CBP plays a role in modulating longevity in this insect model [192]. Of note, the inhibition of another HAT, Gcn5, has been recently observed to positively affect lifespan in yeast and human cells [193].

### 5.3. Histone Phosphorylation and Ubiquitination

Phosphorylation of histone tails has been generally linked to stress-activated gene expression [194,195]. Emerging studies on different model systems indicate that stress responses and longevity might be functionally linked by specific histone phosphorylation. A recent report has highlighted a potential link between nutrient-sensing pathways and chromatin regulation in aging. The study revealed that in yeast, under nutritional stress, there is a specific increase in the level of histone H3 threonine 11 phosphorylation (H3pT11) at stress-responsive genes, which regulates the transcription of genes involved in metabolic transition [196]. In *Drosophila*, mutations mimicking H3S28 phosphorylation have been associated with increased longevity and improved resistance to stress, potentially indicating a role for H3S28 phosphorylation in controlling longevity and stress resistance [197].

Trimethylation of H3K4 and H3K79 using COMPASS and Dot1 methyltransferases, respectively, requires Histone H2B monoubiquitination. A recent study found that H2B monoubiquitination accumulates at the telomere-proximal regions of replicatively aged yeast [198] and that alterations in H2B ubiquitination through the inhibition of the SAGA/SLIK histone deubiquitinase complex affect lifespan in yeast [199]. Very recently, a broader connection has been made linking H2A ubiquitination with the age-dependent accumulation of DNA double-strand breaks, histone proteolysis and genome instability in human models of neurodegeneration [200]. While an increase in H2 ubiquitination seems to be linked to replicative aging, further investigation is needed to understand its role in longevity.

### 5.4. Histone Variants

Alternative histone variants substituting the canonical variants have been functionally associated with the aging process. Histone variants are a crucial component of epigenetic regulation, alongside histone post-translational modifications and chromatin remodeling complexes. These variants possess unique protein sequences and are regulated by specific chaperone systems and chromatin remodeling complexes and can be enriched with specific post-translational modifications to recruit variant-specific interacting proteins to chromatin, thereby imparting distinct character and function to specific regions of chromatin [201]. Among all non-canonical variants, macroH2A, H2A.J, H2A.Z and H3.3 provide very promising observations in aging-related studies. In general lines, an increase in the levels of these variants has been found across different systems, including rats, primates and humans [202]. Specifically, a concomitant increase in H3.3 and decrease in H3.1 (canonical variant) has been observed in aging *Caenorhabditis* and in brain samples from old mice [203,204,205]. Furthermore, the presence of increased levels of H3.3 affects H3 methylation patterns [203]. During aging in human fibroblasts, it was discovered that there was an increase in the expression of H2A.Z and H3.3, while H2A.1 and H3.1 were downregulated [206]. The accumulation of the H2A.J histone variant, which upregulates inflammatory gene expression, was observed in senescent human fibroblasts, as well as in mouse and human epidermal and stem cells with age [207]. Altogether, this suggests that the replacement of canonical histones with histone variants is an epigenetic program characteristic of the aging process. Accordingly, *C. elegans* models lacking histone H3.3 have shorter lifespans and defective transcriptional regulation of lifespan-extending signaling pathways [205].

## 6. DNA Methylation Changes

DNA methylation is mediated by a group of enzymes called DNA methyltransferases (DNMTs), which incorporate methyl groups through a covalent link to the 5th carbon of a cytosine base, generating 5-methylcytosine (5mC). Three different DNMTs have been identified in mammals: DNMT1 (maintenance methylation), DNMT3A and DNMT3B (de novo methylation). DNMTs are crucial for maintaining genomic integrity, and their disruption can lead to chromosome instability and tumor progression [208]. They are required for transcriptional silencing of various sequence classes, and targeted deletion experiments have provided evidence that all three DNMTs are involved in stabilizing the genome, especially repetitive sequences [209,210]. Methylation of DNA is an essential and fine-tuned process during normal development [211]. The presence of these methyl groups decreases the degree of accessibility of the transcription machinery; therefore, its presence or absence definitively impacts the genetic expression [157,212].

In general lines, methylation is commonly localized in regions rich in cytosine-phospho-guanine (CpG) dinucleotides or so-called CpG islands. There is considerable knowledge in the scientific literature indicating a relationship between chronological age and DNA methylation (DNAm) levels. The collective data show that both hyper- and hypo-methylation occur at many CpG sites in the mouse and human genome during the aging process [213,214,215,216,217,218,219]. This highlights that DNAm levels change with age, and the advent of DNA methylation array technology has enabled the identification of specific genomic locations where these changes occur [214,217,218,220,221]. More specifically, several studies have collectively shown that, in the aging context, promoter regions tend to be hypermethylated [140,222,223,224]. Furthermore, this introduces the concept of epigenetic age estimators, which are sets of CpGs that can estimate the age of a DNA source and emphasizes that these estimators not only reflect chronological age but also biological age [223]. DNAm age estimators are highly accurate and are often referred to as epigenetic clocks [225]. Accordingly, age-related DNAm changes seem to occur alongside alterations in DNMT enzyme activities, showing an upregulation of DNMT3B and downregulation of DNMT1/3A in human fibroblasts [226].

It remains currently a challenging task to reveal specific biological functions, which decline or otherwise change as a specific result of altered DNAm patterns in old cells. However, some candidate functions are currently being investigated. Overall, decreased DNA methylation leads to an increase in the accessibility of the genomic DNA, instigating a transformation from heterochromatic to euchromatic regions [227]. Thus, higher genomic instability and risk of movement of retrotransposable elements can be expected [141,228]. Although aging seems to cause casual and tissue-specific changes in DNAm patterns, some specific age-associated epigenetic DNAm modifications have been established through several studies [229,230,231]. Taken together, the correlation between DNA methylation and chronological age is not based on overall genomic entropy but rather on the location of aging-affected CpG sites in either poised promoters (hypermethylated during aging) or strong enhancers (hypomethylated during aging) [232]. The specificity and directionality of these changes in the methylation pattern allow the correlation between aging and DNAm changes to be understood as not a stochastic change but as a mechanism underlying the progression of biological aging [141,223,233]. Accordingly, the study of progeria syndromes, such as Hutchinson–Gilford Progeria and Werner syndrome, has also related abnormal DNAm patterns with accelerated aging [234]. Finally, the consequences of alterations in DNAm patterns during aging might affect the organism’s immune response, as the methylation of immune-related genes has been found to be a driver of the decreased immunocompetence observed in elderly people [235,236]. Additionally, studies with twins have provided very interesting findings regarding patterns of methylation with age. Older monozygotic twins show more divergent patterns than younger twins [237], reinforcing the importance of environmental factors in age-dependent epigenetic alterations.

DNA methylation of the rDNA locus is an especially important aspect of the regulation of rRNA and, consequently, protein synthesis, both in physiological and pathological conditions [238]. An increasing body of evidence indicates that the reduction in protein synthesis is a beneficial response during normal aging, which promotes longevity and can be influenced by nutritional interventions such as caloric restriction (CR). Groundbreaking insights come from experiments in the yeast model, where it has been shown that the number and integrity of rDNA copies directly impact longevity [79] and that the controlled silencing of rDNA or the inhibition of ribosomal gene expression extends lifespan [239,240]. Studies using rodents have shown that there is an increase in the DNA methylation of the rDNA locus as the animals age [241]. This increase was observed in brain, liver, and spleen tissues. In rats, an age-dependent hypermethylation was also reported [242,243]. In human whole blood, no significant relationship between rDNA methylation levels and chronological age was found, but the methylation status of a CpG site in the rDNA promoter was found to be associated with cognitive performance and survival chance [243,244]. Recently, Wang and Lemos confirmed the age-associated hypermethylation of the rDNA locus in mice and developed an epigenetic clock based on DNAm values at the ribosomal locus. This “ribosomal clock” accurately estimates an individual’s age, is sensitive to interventions known to affect lifespan and health, such as CR and can be applied to different species [245].

As for the future of DNA methylation studies, it is important to highlight and reconsider the effect of sex on the observed alterations. As there is an assumption that the alterations in the methylation pattern are equal for both sexes during aging, most of the relevant studies have only been conducted using male organisms. Nevertheless, a study carried out by Masser et al. compared CpG methylation in hippocampal samples of both female and male mice as they age and found that only 5% of the changes correlated with aging are common in both sexes. The majority of aging-related methylation changes were sex-specific, and they were not already pre-established differences [246]. It is very important to further investigate and uncover the true impact this phenomenon has in the process of aging.

## 7. Non-Coding RNAs

The role of non-coding RNAs (ncRNAs) in aging is a more recent development in the field. ncRNAs play critical roles in genetic silencing and epigenetic regulation [247,248]; thus, they are considered here as potential modulators of the aging process via their impact on genomic transcription and the epigenome. Among these, the roles of microRNA (miRNA) and long non-coding RNA (lncRNA) have become highly relevant in past years.

miRNAs are small (19-22nt), non-coding RNAs that play important roles in regulating gene expression by targeting specific mRNAs for degradation or repression of translation [249]. They have been implicated in various biological processes, including development, cell death, and proliferation, and are dysregulated in human diseases, particularly cancer. With over 2000 cataloged human miRNAs, which regulate nearly 60% of all human transcripts, the miRNA regulatory network is extensive and complex, with each miRNA potentially targeting hundreds of transcripts and each transcript being potentially regulated by multiple miRNAs [250]. miRNAs are able to accumulate and interfere with the normal epigenetic regulation of gene expression [251]. Numerous studies have demonstrated that several miRNAs directly impact lifespan by influencing well-established pathways involved in the aging process [252]. These pathways include insulin/insulin-like growth factor (IGF-1) signaling, TOR and translation signaling, sirtuin deacetylases, mitochondrial/ROS signaling, and DNA damage response. These pathways are adaptive mechanisms that maintain organismal homeostasis in response to molecular damage, changes in nutrient availability, and other physiological stressors. Although most of the research on miRNAs and longevity has been conducted on invertebrates, many of the components and functions of miRNA-targeted aging pathways are conserved in mammalian species.

The first discoveries related to the impact of these ncRNAs in aging organisms have been found in *Caenorhabditis* and *Drosophila*, where it was proved that miRNA lin-4 plays an important role in longevity and its loss causes a reduced lifespan [253]. Nevertheless, there is also evidence of some miRNA, which have the opposing effect, acting as pro-aging effectors. Multiple ncRNAs, including lin-4, miR-71, miR-239, and miR-34, have been shown to regulate lifespan in *C. elegans* [254]. These RNAs modulate various pathways such as insulin/IGF-1, DNA damage checkpoint, and autophagy to extend or reduce lifespan [254,255]. The multifunctionality of miR-71 and the negative feedback loop between miR-34 and DAF-16 demonstrate how miRNAs can act as common nodes between different pathways [256]. The role of miRNAs in regulating lifespan in *Drosophila* is complex and sex- and tissue-specific, with let-7 and miR-125 targeting transcription factor Chinmo, involved in neuronal development, to modulate lifespan [257,258]. In contrast to observations in *C. elegans*, loss of miR-34 and miR-100 in *Drosophila* leads to reduced lifespan and early onset neurodegeneration through targeting Eip74EF and VGlut, respectively [259,260]. Only a few lifespan-modulating miRNAs have been identified in mammals. One example is miR-17, which directly extends lifespan in transgenic mice by inhibiting senescence through downstream effectors, including mTOR signaling [261]. In general, miRNAs are downregulated with age [262], and there is evidence that both during normal aging and age-related diseases there is an alteration of miRNA activity in a wide range of model organisms [263,264,265]. More recently, screenings of miRNA signatures in whole-blood samples have identified more than 100 miRNAs that are expressed differentially with aging [266]. Further investigations will be essential to uncover the role of these ncRNA in the aging process and its possible uses as potential therapeutic targets.

Long non-coding RNAs are a diverse group of transcripts that are tissue-specific and show spatio-temporal expression patterns. Mechanisms by which lncRNAs bring about their function include interaction with other RNA species or DNA, scaffolding of subcellular domains or complexes, and regulation of protein activity or abundance. This high specificity makes it a very interesting aging research target [267,268]. Previously, lncRNAs have been associated with processes important for various aging-associated diseases, including cancer, cardiovascular diseases, type II diabetes, and neurodegenerative diseases [269,270,271,272]. Changes in the expression of lncRNAs with aging have been reported in model organisms [273], but the aging-associated changes in the non-coding transcriptome have just started to be extensively studied in humans [274].

## 8. RNA Modifications and Changes in the Epitranscriptome

Epitranscriptomic modifications of RNA, a term encompassing various chemical changes to RNA molecules, have recently emerged as a promising area of aging research. These modifications have the potential to offer new biomarkers and innovative targets for interventions related to longevity and stress resistance. They exert their influence by intricately modulating gene expression through mechanisms like translation initiation and velocity, miRNA interaction, RNA stability, and RNA degradation. However, it is important to note that the precise molecular mechanisms underlying most epitranscriptomic alterations and their associated physiological consequences remain largely uncharted territory, warranting further exploration and understanding. This review focuses on the impact of epigenetic changes occurring in genomic DNA; however, we briefly summarize here the contribution of RNA modifications to the aging process in order to make this overview as complete as possible. The interested reader is referred to excellent recent reviews on the emerging topic of epitranscriptomic alterations of aging [275,276].

Ribosomal RNAs (rRNAs) are heavily modified post-transcriptionally, and some specific modifications are beginning to be mechanistically implicated in the aging process. Examples are dedicated methylases, which introduce specific methyl marks at RNA bases such as m^5^C or m^1^A. One experimental focus was on the yeast m^5^C methyltransferase Rcm1 (NSUN5 in other organisms) and its homologs in various species. These enzymes are responsible for depositing m^5^C modifications at specific positions in ribosomal RNAs, and their loss has been associated with lifespan extension in *C. elegans*, *D. melanogaster*, and *S. cerevisiae* under certain dietary conditions [277]. In *C. elegans*, another m^5^C methyltransferase, the essential nsun-1, mediates a specific 26S rRNA modification. The post-developmental RNA interference (RNAi)-mediated knockdown of nsun-1 in certain genetic backgrounds increased lifespan in a high-throughput screen, and specific depletion of nsun-1 in somatic tissues extended lifespan by about 10% [278]. The improved lifespan and stress resistance resulting from altered rRNA modifications likely stem from changes in translational regulation of gene expression, as most rRNA modifications are located in functional ribosomal regions. Furthermore, a moderate overall reduction in protein synthesis, driven by altered rRNA modifications, may be advantageous and linked to extended lifespan.

While transfer RNA (tRNA) modifications constitute the largest category of RNA modifications, only a very limited number have been observed to extend lifespan upon deletion. Notably, the absence of TRM9 has been associated with an increase in chronological lifespan in yeast [279]. Mammalian cells lacking ALKBH8, a TRM9 ortholog, exhibited reduced translation of selenocysteine proteins and elevated levels of ROS [280]. However, how the modulation of tRNA modifications could impinge on longevity in higher model systems still remains to be elucidated.

Messenger RNAs (mRNAs), apart from being modified by the 5′-cap structure, are also chemically modified internally. m^6^A is the most extensively researched epitranscriptomic mRNA modification due to its dynamic nature, introduced by writers and removed by erasers, with its effects on mRNA translation and stability determined by modification position and interacting reader proteins. In *Drosophila*, the absence of functional Mettl3, an m^6^A writer, as well as the two m^6^A reader proteins YTHDF and YTHDC1, resulted in reduced lifespan, and METTL3 and YTHDF deficiencies were linked to impaired neuronal functions, emphasizing the importance of the m^6^A machinery for neuronal integrity and lifespan in flies [281]. In studies involving mammals, the impact of m^6^A modifying enzymes on aging and senescence varies: one study in long-lived mice demonstrated increased levels of key m^6^A writers and readers, associated with enhanced cap-independent translation, promoting stress resistance and healthy aging [282], while another study found age-related changes in these enzymes, which were influenced by certain lifespan-extending drugs, suggesting a potential role for m^6^A modifications in the mechanisms of these compounds, although a direct causal relationship remains to be established [283].

## 9. Chromatin Organization Changes

Three main aspects of aging-related changes within chromatin organization will be summarized and discussed here: alterations in the distribution of heterochromatin within genomes, the importance of chromatin remodeling complexes, and changes in nucleosome abundance.

### 9.1. Heterochromatin Changes

In eukaryotes, chromatin is classified into two main types, heterochromatin (transcriptionally inactive) and euchromatin (transcriptionally active), based on their structural and functional characteristics [284]. Heterochromatin is further divided into constitutive and facultative heterochromatin, with the former being mainly located in centromeric or telomeric regions containing satellite sequences and transposable elements. The tightly packed constitutive heterochromatic state of these regions helps to protect the genome against harmful chromosomal rearrangements caused by DNA DSB and nonallelic homologous recombination, which can lead to genomic instability, cancers, and hereditary diseases [285]. Among the major contributors to chromatin reorganization, changes affecting histone protein levels and their post-translational modifications are the most relevant.

In general, a holistic loss of heterochromatin, which leads to a decrease in transcriptional silencing, has been described in aged individuals across different model systems [163,286,287,288,289]. Additionally, the loss of heterochromatin affecting the ribosomal DNA has been shown to promote genomic instability and ultimately initiate premature aging in yeast [72,290]. Heterochromatin loss is also observed during senescence, a mechanism used by aging cells to suppress their division to avoid malignancy. Senescent cells show an increase in senescence-associated heterochromatin foci (SAHF), which silence genes that promote cell division [291] but also experience global heterochromatin loss, leading to the expression of genes previously inaccessible to transcription factors, supporting the heterochromatin loss model of aging [286]. Genome-wide heterochromatin loss has been further reported in models of premature aging diseases. For example, cells derived from patients with Hutchinson–Gilford progeria syndrome exhibit chromatin defects that are typically observed during the normal aging process. These defects include loss of heterochromatin, loss of repressive marks, reduced levels of the heterochromatin protein HP1, and enhanced transcription of pericentromeric satellite III repeats [153,292,293]. Additionally, in both chronologically aged human primary mesenchymal stem cells (MSCs) and MSCs from a mouse model of Werner syndrome, decreased expression of the H3K9me3 histone methyltransferase SUV39H1, which regulates heterochromatin formation and maintenance, was observed, leading to abnormalities in the nuclear envelope and a reduction in pericentrometic and telomeric heterochromatin [294,295]. Recently, the Cockayne syndrome causing ATP-dependent chromatin remodeler CSB [296] has been functionally linked to aging-dependent loss of heterochromatin [297].

While it is evident now that aging cells undergo large-scale changes in chromatin structure and accessibility with important consequences for genomic gene expression, it remains challenging to identify specific transcriptional changes as determinants of the aging process. Recent advancements in ChIP-seq, ATAC-seq, and RNA-seq have enabled researchers to study genome-wide chromatin dynamics in small numbers of cells or even single cells. By analyzing chromatin in young and old cells, alterations in gene expression patterns, increased variability in gene expression, and higher cell-to-cell variability have been observed [6]. Sun et al. conducted an initial study on the epigenomic changes in old and young hematopoietic stem cells (HSCs). They found that old HSCs exhibited an increase in H3K4me3 peaks across genes related to HSC identity and self-renewal [149]. Benayoun et al. generated chromatin maps and transcriptomes from multiple tissues and found general patterns in age-related chromatin and transcriptional changes [298]. These alterations may contribute to functional changes in aged HSCs. A related but greatly unexplored question is whether and how an aged chromatin structure modulates the dynamic or sensitivity of gene expression. This might be especially important to understand a possibly impaired activation of gene expression in old cells in response to stress [299,300]. A recent study in yeast shows that replicatively or chronologically aged cells have specific deficiencies in the dynamic response of defense genes upon oxidative stress [301].

Finally, the loss of heterochromatin ultimately leads to the activation of retrotransposable elements, as they are no longer silenced [141], resulting in the disruption of genomic homeostasis [302]. In mammals, an increasing amount of active Alu sequences and LINE-1 repeats have been found in aged organisms. The latter may be connected with the hyperactivation of the immune response system [303].

### 9.2. Chromatin Remodeling

The chromatin landscape and accessibility can be altered through nucleosome assembly, repositioning, and histone variant exchange by ATP-dependent chromatin remodelers, which are categorized into four subfamilies: ISWI, CHD, SWI/SNF, and INO80 [304,305]. Studies in yeast and *Caenorhabditis elegans* have shown that different chromatin remodeling complexes are associated with age-related chromatin remodeling and lifespan regulation. These ATP-dependent chromatin remodeling factors act as positive and negative regulators of longevity by controlling stress resistance. Studies in yeast have shown that deletion of the ISW2 complex leads to changes in nucleosome positioning at multiple gene loci, and this promotes longevity by inducing a genotoxic stress response, similar to the effects of calorie restriction [306]. In contrast, SWI/SNF and Mi2, the catalytic subunit of the nucleosome remodeling and histone deacetylase (NuRD) complex have been identified as positive and negative regulators of longevity in *C. elegans* [307,308], as inactivation of SWI/SNF reduces lifespan and stress resistance mediated by DAF-16/FOXO, while mutations in Mi2 homolog LET-418 extend lifespan and enhance stress resistance in a DAF-16/FOXO-dependent manner. In the same vein, NuRD activity has been found to decline during premature and normal aging in human fibroblasts, affecting nucleosome positioning and increasing the risk of DNA damage [293]. Very recent proteomic studies in mice reveal that the chromatin protein repertoire dynamically changes during different aging stages in an organ-specific manner [309].

### 9.3. Nucleosome and Histone Abundance

Several studies have observed a decrease in the quantity of histones due to a decline in histone protein synthesis as individuals or cells age. Initially described in *Saccharomyces cerevisiae* [310], similar findings in mice and humans have been described [311,312]. The loss of nucleosomes greatly increases chromatin accessibility; therefore, a global upregulation in transcription and increased genome instability can be found in aged cells [140,310]. Consistently, the overexpression of histones in the yeast model leads to an improvement in lifespan [174]. Chen and colleagues conducted a recent study where they reanalyzed H3 chromatin immunoprecipitation and high-throughput sequencing (ChIP-seq) datasets from mouse tissues at various stages of the mouse lifespan. Their findings revealed a series of age-related alterations in H3 occupancy, characterized by both increases and decreases in H3 occupancy. However, the study did not observe significant changes in H3 expression levels [313]. Collectively, the findings from these studies suggest that the loss of histones or nucleosomes during aging is most likely dependent on the specific cell type or context. Of note, a recent study in *Drosophila* and mice suggests that histone protein expression is a functional link between nutrient signaling and longevity via the TOR pathways and, thus, a possible target for lifespan interventions [314].

## 10. Conclusions

The process of aging is a complex biological phenomenon characterized by a multitude of cellular processes. In this review, we have attempted to provide a comprehensive overview of the significant changes that occur at genomes, associated with aging cells and organisms. Table 1 provides a brief summary of recent findings regarding the mechanisms associated with aging described here.

The simultaneous occurrence of these alterations, each with its activating or inhibiting role, affects DNA availability and contributes to genomic instability, overall decline in cellular function, and organismal dysregulation. Consequently, there is an increased susceptibility to age-related diseases such as cancer, cardiovascular diseases, and inflammation. Since the initial suggestion that DNA damage and genome instability are primary drivers and DNA repair is a key factor in aging, followed by the finding that defects in DNA repair can hasten the progression of various age-related diseases, significant progress has been made in understanding the detailed connections between genome instability and every facet of the aging process. Exploring the specific mechanisms by which DNA damage impacts each major process contributing to age-related diseases offers promising avenues to address aging at its fundamental causes, thereby mitigating diseases associated with aging.

It is necessary to conduct further studies to examine how sex, race, and other environmental factors impact these processes. The intricate nature of aging in vertebrates and the interconnections between its various hallmarks indicate that much remains to be unraveled. The utilization of new experimental models and research techniques will provide crucial insights into the underlying mechanisms. Consequently, the importance of efficient genome surveillance has been reported in extremely long-lived invertebrate and mammalian species [315,316,317]. Significant advancements have been made in the field of aging over the past decade with respect to causative genomic alterations, and with the advent of next-generation sequencing methods, it is now feasible to elucidate the underlying mechanisms of aging in greater detail. Single-cell sequencing and transcriptome profiling emerge as valuable tools to comprehend the true impact of epigenetic alterations on cells and their physiological consequences. These techniques enable early interventions to mitigate the adverse effects of aging and age-related diseases on organisms. Interventions based on a reinforcement of DNA repair and genome surveillance aimed at slowing down the onset of aging might be difficult to achieve given the complexity of the biological processes. However, very recent advances in the yeast model demonstrate that genetic rewiring in favor of keeping the balance between genomic stability and mitochondrial function is able to postpone significantly the entry into aging [318], which might open the door to similar genetic interventions in higher systems.

## Figures and Tables

**Figure 1 ijms-24-14279-f001:**
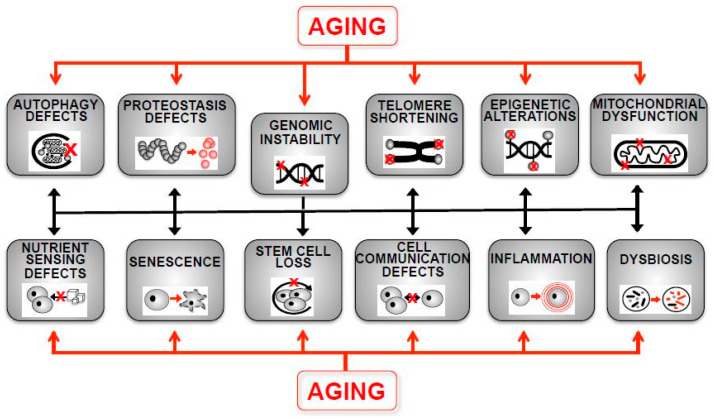
The hallmarks of aging and the central role of genomic instability. Indicated are the cellular and organismal functions, which are known to contribute to the aging process (adapted from López-Otín et al. [5]). Genomic instability is functionally interconnected with all other hallmarks because it comprises telomere length shortening, causes epigenetic alterations through the mutation of epigenetic modifiers or during the process of DNA repair, alters proteostasis via the increase in aberrant, mutated proteins, affects macro-autophagy via its involvement in the DNA repair process, leads to mitochondrial dysfunction via mutations in mitochondrial genomes, can trigger senescence upon DNA damage, is dependent on nutrient sensing pathways such as TOR for DNA repair, might alter intercellular communication through the impairment of the activation of the relevant signaling pathways, lowers the stem cell renewal potential via epigenetic alterations, triggers inflammation programs via DNA mutation and damage and might induce dysbiosis through the accumulation of genomic mutations in intestinal cells.

**Table 1 ijms-24-14279-t001:** The impact of genomic instability on longevity: Overview of selected recently reported mechanisms and consequences *.

Hallmark	Alteration	Consequence	Reference
Telomere shortening	Telomere attrition	Decreased lifespan	[31,40]
	Mitochondrial dysfunction, Accumulation of ROS	Increase in senescence	[33,34]
DNA Damage and Repair	rDNA damage accumulation	Increased genomic instability, decreased lifespan	[74,75]
	Decline of DDR capacity: Upregulation of alt-NHEJ repair	Increased mutagenic alterations, increased genomic instability, decreased lifespan.	[115,116]
Histone modification	Loss of repressing marks	Loss of heterochromatin: increased genomic instability, aberrant gene expression	[138]
	Increase in H3K4me3 marks	Activation of aging-related genes	[145]
	Absence of H3K9 deacetylase	Increased genome instability: reduced lifespan	[180]
	Increased H2A ubiquitination	Increased histone proteolysis, DNA DSB and genome instability	[200]
DNA methylation	Hypermethylation of promoter regions	Degradation of transcriptional networks, genomic instability, decreased lifespan	[224]
	Hypermethylation of rDNA	Ribosomal dysregulation, metabolic control during aging	[244]
Chromatin changes	Increased loss of heterochromatin	Increased loss of transcriptional silencing, genomic instability, accelerated aging	[289]
	Altered chromatin protein repertoire	Altered chromatin and gene expression dynamics	[309]
ncRNAs	Differential miRNA expression patterns	Age-related tuning of protein expression	[266]

* Articles published in the past five years (2018–2023).

## Data Availability

Not applicable, no new data were created during the present study.

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
