# Peer review of "Genomic Instability and Epigenetic Changes during Aging"

_ijms, 2023, doi:10.3390/ijms241814279_

Round 1
Reviewer 1 Report
The review article titled " Genomic instability and epigenetic changes during aging " is undeniably well-written, offering a comprehensive overview of the current state of knowledge regarding the role of genomic instability in the aging process.
However, one notable drawback is the absence of a dedicated chapter on changes in RNA modifications during aging. Given the growing interest in the link between RNA modifications and age-related processes, a section addressing this topic would have further enriched the article's content and relevance.
Furthermore, the review relies heavily on secondary sources, with a considerable number of references (302) being review articles. While reviews serve as useful tools to synthesize existing knowledge, the lack of citation of primary articles diminishes the article's potential to provide readers with the most current and rigorous scientific findings.
Minor:
Lines 64-66: Here, I am missing information about RNA and DNA epigenetic modifications. Additionally, citing recent literature sources would be helpful.
Author Response
Thank you for your comments. We have revised the manuscript.
Reviewer 2 Report
This is a well-constructed and thorough review that summarizes the current state of knowledge within the field. With some minor changes, the manuscript could be improved:
1. On page 3, an expanded discussion of mitochondrial dysfunction, and what drives it, would be helpful in the context of ROS and telomere shortening.
2. Please elaborate on how dysfunction in DNA repair specifically underlies various progerias.
3. In section 5, if possible, a figure depicting the major histone modifications would provide greater clarity for the section.
Author Response

(The authors gave the same response as above.)
